# Optimizing Resistance Training Technique to Maximize Muscle Hypertrophy: A Narrative Review

**DOI:** 10.3390/jfmk9010009

**Published:** 2023-12-29

**Authors:** Patroklos Androulakis Korakakis, Milo Wolf, Max Coleman, Ryan Burke, Alec Piñero, Jeff Nippard, Brad J. Schoenfeld

**Affiliations:** 1Applied Muscle Development Laboratory, Department of Exercise Science and Recreation, CUNY Lehman College, Bronx, NY 10468, USA; polkarots@gmail.com (P.A.K.); milowolf@outlook.com (M.W.); colemanmax888@gmail.com (M.C.); alec.pinero@gmail.com (A.P.); 2STRCNG Incorporated OA Jeff Nippard Fitness, Oakville, ON L6L 1W4, Canada; jeffnippard@gmail.com

**Keywords:** resistance training form, muscle development, muscle growth

## Abstract

Regimented resistance training (RT) has been shown to promote increases in muscle size. When engaging in RT, practitioners often emphasize the importance of appropriate exercise technique, especially when trying to maximize training adaptations (e.g., hypertrophy). This narrative review aims to synthesize existing evidence on what constitutes proper RT exercise technique for maximizing muscle hypertrophy, focusing on variables such as exercise-specific kinematics, contraction type, repetition tempo, and range of motion (ROM). We recommend that when trying to maximize hypertrophy, one should employ a ROM that emphasizes training at long muscle lengths while also employing a repetition tempo between 2 and 8 s. More research is needed to determine whether manipulating the duration of either the eccentric or concentric phase further enhances hypertrophy. Guidelines for body positioning and movement patterns are generally based on implied theory from applied anatomy and biomechanics. However, existing research on the impact of manipulating these aspects of exercise technique and their effect on hypertrophy is limited; it is therefore suggested that universal exercise-specific kinematic guidelines are followed and adopted in accordance with the above recommendations. Future research should investigate the impact of stricter versus more lenient exercise technique variations on hypertrophy.

## 1. Introduction

Regimented resistance training (RT) has been shown to promote increases in muscle size (also known as hypertrophy) [1]. Maximizing muscle hypertrophy requires the manipulation of certain RT variables, such as training volume (i.e., sets performed per muscle group per week), intensity of effort (i.e., proximity to momentary muscular failure), and range of motion (ROM) [2]. Training technique is another variable often referred to as a key component of effective RT. The term “training technique” is often used to describe the different variables that make up an exercise, and it has been proposed that proper technique may enhance muscle development [3].

Currently, there is no universally agreed-upon definition in the scientific literature as to what constitutes proper technique and its constituent components. The National Strength and Conditioning Association (NSCA) Exercise Technique Manual [4] describes exercise technique components as a checklist that includes the following variables: primary muscle groups involved, correct grip width/orientation, stance, body position, and ROM. However, it is worth noting that what constitutes “appropriate technique” can be heavily affected by an individual’s specific training goals, with different goals potentially requiring a specific technique to maximize adaptations (e.g., hypertrophy versus power training) [5]. Thus, from a muscle hypertrophy standpoint, we suggest the following definition:

Resistance training exercise technique pertains to the controlled execution of bodily movements to ensure an exercise effectively targets specific muscle groups while minimizing the risk of injury. This involves the orchestration of body positioning and alignment, ROM, and repetition tempo.

When prescribing RT in both the literature and in practice, there is an emphasis on the importance of appropriate exercise technique to enhance the effectiveness of RT as well as to potentially prevent injury, although research directly exploring either outcome is scarce [6]. As it stands, guidelines for the prescription of exercise technique include exercise-specific body positioning, alignment, and ROM recommendations, as well as universal tempo recommendations that vary depending on individual training goals.

Researchers have highlighted the importance of learning “proper technique” in the early stages of engaging in RT [7]. However, although it can be inferred that the authors are referring to some of the variables mentioned above, they do not explicitly define what constitutes proper technique, creating ambiguity in their recommendations. Similarly, previous studies examining the manipulation of variables to maximize muscle hypertrophy often define “proper form” [8] as the guidelines outlined in the methods section of this study itself. Although one can extrapolate what constitutes optimal training technique by reading the literature on various variables comprising an RT exercise, no scholarly paper to date has endeavored to synthesize the current literature to develop guidelines for proper technique to maximize muscle hypertrophy, including studies that have specifically focused on the manipulation of different variables to maximize muscle hypertrophy [9,10]. Therefore, this review aims to synthesize the evidence on what constitutes an optimal approach to training techniques for maximizing hypertrophy and provide recommendations for future research on the effect of the RT technique on muscle hypertrophy.

## 2. Repetition Tempo

Repetition tempo is one of the main components often mentioned when discussing exercise techniques to maximize hypertrophy. The conventional RT technique usually involves the inclusion of combined eccentric and concentric actions during each repetition, and it has been proposed that both eccentric and concentric actions should be employed in order to maximize muscle hypertrophy [11]. It is common for exercise professionals to emphasize the importance of either performing the concentric or eccentric phase of repetition with a specific duration [12]. Some researchers have even proposed performing both actions in a “super slow” manner, but the benefits of this strategy on hypertrophy remain poorly supported in the literature, and some evidence indicates a detrimental effect of very slow tempos [13].

A systematic review and meta-analysis by Schoenfeld et al. [11] found that a wide range of repetition durations (0.5 to 8 s) resulted in similar hypertrophy. However, the authors noted that their review did not specifically analyze whether different eccentric and concentric tempo duration combinations could lead to different hypertrophic outcomes. Notwithstanding, their findings did highlight that repetition tempo may not be as critical of a component for hypertrophy adaptations as previously hypothesized.

More recently, Wilk et al. [14] reviewed the influence of tempo on hypertrophy and strength adaptations and found that a combination of slower eccentric and faster concentric repetition seems to be best for maximizing muscular development. However, given the limited literature on the topic and the heterogeneity among the protocols of the studies reviewed, the authors could not provide a specific tempo recommendation for each repetition phase and simply suggested a duration of ~3 to ~8 s per repetition.

The recommendation by Wilk et al. [14] for using a slower eccentric and faster concentric tempo to maximize hypertrophy seems to stem mostly from the studies by Keeler et al. [15] and Nogueira et al. [16]. Keeler et al. [15] found that when the duration of the eccentric phase was similar among conditions (5 vs. 4 s), prolonging the duration of the concentric phase (10 s) did not increase lean body mass (LBM) when compared to a faster concentric phase (2 s). It is important to note that Keeler et al. [15] did not directly measure hypertrophy changes but rather assessed body composition changes via air displacement plethysmography (Bod Pod); thus, these findings should be interpreted cautiously. Additionally, Keeler et al. [15] noted that there were no significant pre-to-post LBM changes in either of the groups, specifically stating that the significant changes in strength and aerobic capacity were observed “in the absence of changes in LBM”. In contrast, Nogueira et al. [16] demonstrated that a faster concentric phase duration (1 s) resulted in significantly greater biceps brachii and rectus femoris hypertrophy when compared to a slower concentric phase duration in a cohort of older men (3 s).

Some research indicates that extending the eccentric phase may enhance the RT-induced hypertrophic response. Pereira et al. [17] found that performing a 4 s eccentric elicited greater absolute biceps brachii hypertrophy when compared to a group performing a 1 s eccentric, with both groups employing a 1 s concentric repetition tempo. It should be noted that the findings of the study by Pereira et al. [17] showed no statistically significant differences in hypertrophy between the two groups, but the effect sizes favored the group performing the extended eccentric phase. A more recent study [18] looked at the effect of different eccentric repetition durations (2 vs. 4 s) during repetitions that employed both concentric and eccentric actions on lower limb hypertrophy and strength and found that, although overall lower limb hypertrophy was similar, the group that performed longer eccentric repetitions experienced greater increases in vastus medialis muscle thickness. Contrastingly, Pearson et al. [19] found that a 1 s eccentric resulted in marginally greater quadriceps muscle thickness increases compared to a 3 s eccentric in trained men performing the leg extension. Alternatively, Shibata et al. [20] explored the effects of prolonging the eccentric phase during parallel squatting on hypertrophy and found no differences in thigh cross-sectional area changes between a 2 s-eccentric and a 4 s-eccentric group, with both groups employing a 2 s concentric tempo. Lastly, Gillies et al. [21] compared a group performing all exercises using a 6 s eccentric and a 2 s concentric to a group performing all exercises using a 6 s concentric and 2 s eccentric. They found that the 6 s concentric group experienced significantly greater increases in vastus lateralis type I and IIA fiber area, while the 6 s eccentric group experienced increases only in vastus lateralis type 1 fiber area. The conflicting findings highlight the uncertainty on the topic and preclude the ability to draw strong conclusions for practical application.

Overall, it does indeed appear that significant increases in hypertrophy can occur with repetition durations between 2 and 8 s, allowing for a plethora of acceptable eccentric and concentric tempos. Based on the current literature, it is unclear whether extending the concentric or eccentric phase of a repetition will lead to greater hypertrophy, although it is advisable that eccentric actions be performed in a sufficiently controlled manner to ensure that the muscle controls the descent of the weight rather than relying solely on gravitational forces.

## 3. Range of Motion

Range of motion is defined as the degree of movement that occurs at a specific joint during the execution of an exercise [22]. Traditionally, a full ROM, herein defined as the largest exercise-specific degree of ROM that can be achieved at each joint, has been recommended for maximizing muscle hypertrophy [12]. This recommendation is consistent with systematic reviews and meta-analyses that indicate a hypertrophic benefit to utilizing a full ROM over a partial ROM [23,24]. However, these papers dichotomized ROM as either full or partial ROM, without distinguishing the influence of muscle length. Interestingly, some literature suggests the ROM used may influence regional hypertrophy, with training at long muscle lengths eliciting more distal hypertrophy than training at short muscle lengths [25]. A more recent ROM systematic review and meta-analysis by Wolf et al. [22] found that a partial ROM performed at longer muscle lengths was potentially superior to a full ROM for hypertrophy. Similarly, a systematic review by Kassiano et al. [26] also found that a partial ROM at long muscle lengths is sufficient to promote optimal muscle growth for muscles such as the quadriceps femoris, biceps brachii, and triceps brachii. That said, caution must be taken when interpreting these results, as only three studies were included in the subgroup analysis by Wolf et al. [22].

Since the publication of the Wolf et al. [22] meta-analysis and the Kassiano et al. [26] systematic review, two additional studies have compared longer-muscle-length partial ROM to full ROM training for hypertrophy. Kassiano et al. [27] assessed hypertrophy of the gastrocnemius when performing ankle plantarflexion RT with either a full ROM, shorter-muscle length partial ROM (50% of full ROM), or longer-muscle length partial ROM (50% of full ROM). Results showed that the medial gastrocnemius grew significantly more in the longer-muscle-length partial ROM group than both the full ROM and shorter-muscle-length partial ROM groups. Alternatively, the lateral gastrocnemius grew significantly more in both the full ROM and longer-muscle-length partial ROM groups vs. the shorter-muscle-length partial ROM group. Overall, this study lends further credence to using longer-muscle-length partial ROM to maximize muscle hypertrophy. An as-yet unpublished study comparing full ROM to longer-muscle length partials has been presented as a conference abstract [28]. Based on the abstract, hypertrophy of the hamstrings, gluteus maximus, and adductor muscles was compared when performing either full ROM or longer-muscle-length partial ROM (50% of full ROM) on the multi-hip machine. Generally, hypertrophy was superior in the longer-muscle-length partial ROM group, which saw significantly greater growth of the hip extensors as a composite and, more specifically, the gluteus maximus and long head of the biceps femoris. The differences between the semitendinosus and semimembranosus were not significantly different, but effect sizes favored the partial ROM group. It should be emphasized that this study methods have not been published, and thus the results must be interpreted cautiously.

Based on the current literature, it appears that utilizing a ROM that biases longer muscle lengths should be the default approach to exercise technique when trying to maximize hypertrophy. Although traversing the end ROM at shorter lengths may not promote an added benefit, and perhaps stopping degrees short could conceivably enhance hypertrophy via different mechanisms [29], more research on different ROM configurations is needed to draw stronger conclusions on the topic. Additionally, the current literature on long-muscle-length training is limited to a few muscles, and thus more research is needed to understand whether some muscles may be more predisposed to longer-length training than others.

## 4. Exercise-Specific Kinematics

Exercise-specific technique guidelines are meant to delineate the “correct” performance for a given exercise. From a body positioning and movement pattern standpoint, the current guidelines are a product of years of technique refinement dating back to the exercises’ inception. The majority of RT exercises were not created based on scientific research that looked at different kinematic configurations of an exercise to find which configuration is better for optimizing hypertrophy or strength. Instead, RT exercise-specific guidelines specific to body positioning and movement patterns were based on a combination of extrapolation of applied anatomy concepts and biomechanical principles.

Exercise-specific instructions on body alignment and positioning (grip width, foot positioning, bar placement) intend to make an exercise biomechanically efficient, effective, and safe for the target muscle(s). For example, the NSCA Exercise Technique Manual for Resistance Training [4] provides a detailed list of descriptions for performing the barbell back squat exercise correctly. Guidelines include instructions like “ensure the weight is evenly distributed between the heels and midfoot [4] when performing a barbell back squat to avoid falling”, as well as muscle and joint-specific instructions on completing the exercise successfully (e.g., “extend the hips and knees fully”). The guidelines also highlight actions that must be avoided to adhere to the correct exercise technique (e.g., “do not allow the knees to shift inward or outward” [4].

Although some of the exercise-specific guidelines are based on biomechanical reasoning (e.g., keeping the bar close to your body when performing a barbell deadlift to optimize the moment arms at the knee and hip joints), objective research directly exploring the effect of different exercise-specific techniques on hypertrophy is limited. One of the very few direct studies on the topic explored the effect of altering foot position during calf training over a 9-week study period [30]. This study found that training with different foot positions, i.e., foot pointed outward or inward, resulted in greater hypertrophy of different aspects of the calf muscles. Other studies have explored the effect of different foot positioning or grip width on muscle activation on exercises such as the machine leg press, lat pull-down, and barbell bench press [31,32,33]. However, given that muscle activation is not necessarily a good predictor of hypertrophy [34,35], the practical implications of these findings for exercise technique guidelines remain questionable.

Additionally, previous literature has attempted to explore the biomechanics of specific exercises in an attempt to provide recommendations for manipulating different biomechanical components to optimize exercise performance and safety. Schoenfeld [36] examined the kinematics and kinetics involved in squatting exercises and provided a list of practical recommendations pertaining to ROM, squatting stance width, bar placement, etc. It is important to note that the recommendations provided in the review paper by Schoenfeld [36] pertain mostly to squatting performance and safety and do not explore the potential influence of biomechanical alterations on hypertrophy. Moreover, commonly emphasized technique guidelines pertaining to the reduction of injury risk when performing RT, such as associations between lumbar spine flexion and lower back pain, are often based on mechanistic evidence, including human and animal cadaver studies [37,38], rather than longitudinal or cross-sectional evidence in living humans [39]. Thus, conclusions from the studies must be interpreted cautiously.

When trying to optimize repetition technique, emphasis is often placed on the importance of avoiding the involvement of ancillary muscle groups, especially on exercises meant to isolate a specific muscle. In practice, exercise technique is often categorized accordingly as strict or non-strict. Strict technique refers to an approach that effectively directs maximum stimulation to the target muscle group(s) by minimizing the direct involvement of other muscle groups, while non-strict technique allows for the involvement of ancillary muscle groups. For example, on a barbell biceps curl, the use of strict technique would involve a more upright posture with minimal hip and leg drive, while non-strict technique might involve a variable posture that sways back and forth, permitting assistance from the gluteals, erector spinae, and other muscles in addition to the biceps brachii.

To the authors’ knowledge, only one previous study has attempted to explore the topic of strict versus non-strict techniques, albeit indirectly. Arandjelovic [40] employed mechanical modeling/simulation to explore the relationship between external momentum and muscular force in the lateral raise exercise to determine how muscular force may change when external momentum is supplied at the beginning of each repetition. The author concluded that the use of a moderate amount of momentum at the beginning of each repetition in a set can allow for the use of heavier loads and thus “a better overload” of muscles in biomechanically advantageous positions, which conceivably could enhance muscular adaptations. Conversely, more excessive amounts of momentum resulted in lower loading of the target muscles, thus potentially impairing the hypertrophic stimulus of an exercise. However, given the current state of the literature in regard to the effect of ROM on hypertrophy, it is possible that the use of moderate external momentum at the beginning of each repetition could potentially have deleterious effects by decreasing time spent at long muscle lengths on some exercises. Alternatively, the use of external momentum may allow for more time spent training at long muscle lengths for other exercises (e.g., bent over a barbell row), thus resulting in greater hypertrophy. The findings of this simulation paper must be viewed with circumspection, given that it did not directly assess changes in muscle mass over time.

As it stands, there is no literature directly examining the effect of strict versus non-strict repetition techniques on hypertrophy. While it could be argued that the involvement of other muscles may negatively affect the hypertrophic stimulus imposed on the target muscle, it is currently unclear whether that is truly the case. Although basing technique guidelines on biomechanical principles has a logical basis, it remains unclear how strict the form needs to be to optimize hypertrophic outcomes while maintaining adequate safety standards. In the absence of direct evidence on the topic, it seems advisable to adopt current exercise-specific guidelines as they relate to body positioning and movement patterns. Moreover, differences in anthropometry between individuals will necessarily require some alteration in kinematics to facilitate safe and effective performance. Further research is needed to provide greater clarity on the importance of strict body positioning and movement pattern recommendations for optimizing muscle hypertrophy.

## 5. Practical Recommendations

Based on the current available evidence, we recommend that exercise technique in RT programs designed to maximize muscle hypertrophy should employ a ROM that allows a muscle to be fully stretched while utilizing an eccentric and concentric phase duration that spans an overall repetition duration of 2–8 s (see Table 1). This recommendation allows for a broad range of concentric and eccentric durations. For example, a 1 s concentric and 2 s eccentric, a 4 s concentric and 4 s eccentric, or a 7 s concentric and 1 s eccentric would all technically fit within these guidelines. Existing research on the impact of manipulating body position and movement patterns and their effect on hypertrophy is limited, and it is therefore suggested that universal exercise-specific kinematic guidelines be incorporated into the above recommendations.

It is currently unclear as to whether a less strict/lenient technique, via the introduction of movement generated by the involvement of unintended muscle groups, will yield inferior results to stricter repetitions that minimize the involvement of unwanted muscle groups, especially when the ROM and tempo recommendations are followed. As long as one can take the intended muscle group to momentary failure, the involvement of other muscle groups may not play a significant role in impairing muscle hypertrophy.

However, it may be that if the other muscle groups involved are working primarily at short muscle lengths (e.g., the hip extensors during a bicep curl), the fatigue generated by their involvement may not be worth the trade-off as they are unlikely to benefit from the stimulus provided. Indeed, they would be very far from reaching momentary failure, and partial repetitions at shorter-muscle lengths are not as hypertrophic as partial repetitions at longer-muscle lengths [22]. Notwithstanding, it is possible that the fatigue generated may be negligible without any meaningful impact on muscle recovery [41].

On the contrary, it could also be argued that if the assisting muscles are at longer muscle lengths (e.g., the hip extensors during the bent over row), they may also experience increases in muscle hypertrophy since sets performed far from momentary failure (e.g., 5–8 repetitions in reserve) can also lead to hypertrophy increases [42]. However, given the paucity of data on the subject, it is recommended to minimize the involvement of other joints to ensure repetitions adhere to the recommended repetition tempo and ROM.

## 6. Conclusions

The repetition technique to maximize muscle growth may be more flexible than previously thought, with the main variable of influence appearing to be ROM, specifically emphasizing training at long muscle lengths. Given the current literature, eccentric and concentric repetition tempos can vary based on preference, as long as the total repetition length is between 2 and 8 s.

Future research should attempt to further explore the concept of what constitutes appropriate RT technique for maximizing hypertrophy and the different perceptions as well as practices that exist among exercise professionals, physique sport athletes, and coaches. Additionally, future research should also directly explore the effect of strict versus more ‘lenient’ exercise techniques on hypertrophy, specifically focusing on whether the contribution of non-targeted muscle groups affects hypertrophy outcomes in exercises intended to target a specific muscle.

## Figures and Tables

**Table 1 jfmk-09-00009-t001:** Technique recommendations to maximize muscle hypertrophy.

Variable	Evidence	Maximization Recommendation
Tempo	Moderate	A repetition tempo of 2–8 s seems to be sufficient to maximize hypertrophy, and it is currently unclear whether extending the concentric or eccentric phase of a repetition will lead to greater hypertrophy
ROM	Moderate	Employ a ROM that allows for muscles to be fully stretched
Involvement of non-target muscles	N/A	Diminish involvement by minimizing the use of external momentum when possible

N/A, not applicable.

## Data Availability

Not applicable.

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
