# Peer review of "Optimizing Resistance Training Technique to Maximize Muscle Hypertrophy: A Narrative Review"

_jfmk, 2023, doi:10.3390/jfmk9010009_

Round 1
Reviewer 1 Report
Comments and Suggestions for Authors
It is appropriate to publish it in present form. I do not have any suggestions.
Author Response
Thank you for the review. We appreciate your time in reviewing.
Reviewer 2 Report
Comments and Suggestions for Authors
The manuscript is well written and deals with an interesting and often overlooked area of resistance training. The reviewer has only a few suggestions regarding the manuscript:
line 138: the last part of the sentence is a little awkward and should be rephrased ......"the muscle lowers the weight as opposed to gravity"
from line 140, section under "Range of motion" subheading- it would be worth including some discussion, perhaps a few lines, about regional, nonuniform muscle hypertrophy and whether it may be related to ROM during resistance training.
from line 183 to about 234 section under "Exercise-specific kinematics" subheading- this section is a little superfluous and should be more concise, as some of this information is not directly related to the content and doesn't seem to be very relevant to it.
Author Response
The manuscript is well written and deals with an interesting and often overlooked area of resistance training. The reviewer has only a few suggestions regarding the manuscript. |
We thank you for your kind words. |
line 138: the last part of the sentence is a little awkward and should be rephrased ......"the muscle lowers the weight as opposed to gravity" |
We have rephrased the highlighted sentence. |
from line 140, section under "Range of motion" subheading- it would be worth including some discussion, perhaps a few lines, about regional, nonuniform muscle hypertrophy and whether it may be related to ROM during resistance training. |
Excellent suggestion, we have added a few lines discussion the effect of ROM on regional hypertrophy. |
from line 183 to about 234 section under "Exercise-specific kinematics" subheading- this section is a little superfluous and should be more concise, as some of this information is not directly related to the content and doesn't seem to be very relevant to it. |
We have modified the “Exercise-specific kinematics” section to be much more concise by removing the initial paragraphs that described the evolution of the barbell back squat. |
Reviewer 3 Report
Comments and Suggestions for Authors
The authors present an interesting literature review on a much-discussed topic in gym and training places.
- It would be interesting to combine the literature review with a possible molecular-level explanation of the proposed results
- The advice is interesting but a little generic; it would be better to also have some practical examples, even of a possible workout
- Do you think that even a session on suspension training could be included, given today's feedback, perhaps for particular categories such as the elderly (10.3389/fspor.2022.950949)
- Proposals on possible future studies would be interesting to characterize the response to RT better
Comments on the Quality of English Languagejust few
Author Response
The authors present an interesting literature review on a much-discussed topic in gym and training places. |
We thank you for your efforts in reviewing our manuscript. |
The advice is interesting but a little generic; it would be better to also have some practical examples, even of a possible workout |
We have added some practical examples in our manuscript |
Proposals on possible future studies would be interesting to characterize the response to RT better |
The manuscript includes a plethora of suggestions for future studies on RT training technique |
Reviewer 4 Report
Comments and Suggestions for Authors
Dear authors,
I reviewed your manuscript with great interest. The scientific perspective often ignores resistance training techniques. As you reported, the existing research on the impact of manipulating these aspects of exercise technique and their effect on hypertrophy is limited.
I have just a couple of suggestions to enhance the quality of your manuscript.
Firstly, there is a conceptual problem: if the existing research on the topic is limited, how can you provide an optimal approach to training techniques?
Talking about range of motion, the results of the recent systematic review of the effects of range of motion on muscle hypertrophy are lacking (DOI: 10.1519/JSC.0000000000004415); include them in your paragraph.
As you mentioned, your review intends to consolidate current knowledge on the optimal exercise method for achieving maximum muscle growth during resistance training. On the other hand, taking into consideration the fact that a great number of publications on the optimisation of resistance training have recently been published, it is not entirely evident how your article contributes to the advancement of the state of the art. Increase the justification of your paper and concentrate on the existing knowledge and areas that need to be explored to encourage more research.
Author Response
I reviewed your manuscript with great interest. The scientific perspective often ignores resistance training techniques. As you reported, the existing research on the impact of manipulating these aspects of exercise technique and their effect on hypertrophy is limited. |
We thank you for your review and appreciate your interest in the topic of our manuscript. |
Firstly, there is a conceptual problem: if the existing research on the topic is limited, how can you provide an optimal approach to training techniques? |
Although the research that directly examines exercise technique is limited, the literature on some of the main components (eg: ROM, tempo) of exercise technique is enough to provide evidence-based guidelines for how an individual can manipulate said components to maximize muscle hypertrophy. |
Talking about range of motion, the results of the recent systematic review of the effects of range of motion on muscle hypertrophy are lacking (DOI: 10.1519/JSC.0000000000004415); include them in your paragraph. |
We revised our manuscript to and have now included the Kassiano et al (2023) systematic review in our discussion. |
As you mentioned, your review intends to consolidate current knowledge on the optimal exercise method for achieving maximum muscle growth during resistance training. On the other hand, taking into consideration the fact that a great number of publications on the optimisation of resistance training have recently been published, it is not entirely evident how your article contributes to the advancement of the state of the art. Increase the justification of your paper and concentrate on the existing knowledge and areas that need to be explored to encourage more research. |
Thank you for your excellent suggestion. We have increased our justification of our review.
Our review aims to synthesize the current available evidence specifically on exercise technique for optimizing hypertrophy versus focusing on the different variables that need manipulation to optimize muscle hypertrophy (eg: training volume, intensity, rest periods etc). In addition to our review being the first to directly synthesize the current scientific evidence on manipulating the different components of exercise technique to maximize hypertrophy, we also provide a definition for exercise technique as well as highlight the lack of evidence in regards to exercise-specific kinematic modifications and their effect on hypertrophy.
Additionally, we also highlight the need for more research on the effect of strict versus lenient technique on hypertrophy, including highlighting the need for research on better understanding what constitutes “strict” exercise technique as it pertains to muscle hypertrophy.
The current available literature on optimizing resistance training variables for hypertrophy does not directly examine, or sometimes even mention, training technique, let alone all the components that comprise it. For example, the systematic review on maximizing muscle hypertrophy by Krzysztofik et al (2019) does not mention exercise technique, ROM or exercise-specific kinematics. Similarly, the umbrella review by Bernardez-Vazquez et al (2022) also does not make any references to technique, ROM or exercise-specific kinematics. Although the umbrella review by Bernardez-Vazquez et al does briefly discuss repetition tempo, there is no discussion on specifically manipulating the duration of the eccentric or concentric phases of a repetition and how that may affect hypertrophy, something that we often see being advocated in practice and even in personal training resources (eg: NASM Essentials of Personal Training) |
Round 2
Reviewer 3 Report
Comments and Suggestions for Authors
Even if the authors did not answer all my suggestions, the ms could be published.
- It would be interesting to combine the literature review with a possible molecular-level explanation of the proposed results
- Do you think that even a session on suspension training could be included, given today's feedback, perhaps for particular categories such as the elderly (10.3389/fspor.2022.950949)
Author Response
Thank you again for your efforts in reviewing our manuscript. We appreciate the feedback, but believe the current form of our manuscript satisfies all comments made that are relevant to the scope of the topic.
Reviewer 4 Report
Comments and Suggestions for Authors
well done
Author Response
Thank you!